# A High-Efficiency Radio-Frequency-Assisted Hot-Air Drying Method for the Production of Restructured Bitter Melon and Apple Chips

**DOI:** 10.3390/foods13020197

**Published:** 2024-01-08

**Authors:** Wei Jin, Min Zhang, Arun S. Mujumdar

**Affiliations:** 1State Key Laboratory of Food Science and Resources, Jiangnan University, Wuxi 214122, China; jinwei_jnu@163.com; 2Jiangsu Province International Joint Laboratory on Fresh Food Smart Processing and Quality Monitoring, Jiangnan University, Wuxi 214122, China; 3China General Chamber of Commerce Key Laboratory on Fresh Food Processing & Preservation, Jiangnan University, Wuxi 214122, China; 4Department of Bioresource Engineering, Macdonald Campus, McGill University, Sainte-Anne-de-Bellevue, QC H9X 3V9, Canada; arunmujumdar123@gmail.com

**Keywords:** restructured bitter melon and apple chips, radio-frequency-assisted hot-air drying (RFHAD), exhaust air recirculation (EAR), drying characteristics, quality attributes

## Abstract

Nowadays, consumers are increasingly demanding processed food products with high levels of beneficial components. Bitter melon and apple are both nutritious foods rich in bioactive compounds. In this study, restructured bitter melon and apple chips were processed using four drying techniques: hot-air drying with/without exhaust air recirculation (EAR), and radio-frequency-assisted hot-air drying (RFHAD) with/without EAR. The drying characteristics, effective moisture diffusivity (D_eff_), specific energy consumption (SEC), total energy consumption (TEC), and some selected quality characteristics of the dehydrated chips were evaluated. The experimental results show that the application of radio frequency (RF) energy significantly facilitates water evaporation in the drying material, resulting in a significant (*p* < 0.05) reduction of drying duration by 31~39% over the experimental test parameters. The higher D_eff_ values obtained from RFHAD and RFHAD + EAR were 6.062 × 10^−9^ to 6.889 × 10^−9^ m^2^/s, while lower SEC values ranged from 301.57 to 328.79 kW·h/kg. Furthermore, the dried products possessed better or fairly good quality (such as a lower color difference of 5.41~6.52, a lower shrinkage ratio of 18.24~19.13%, better antioxidant capacity, higher chlorophyll, total flavonoid, and total phenolic content, a lower polyphenol oxidase activity of 49.82~52.04 U·min^−1^g^−1^, smaller diameter and thickness changes, and a lower hardness of 27.75~30.48 N) compared to those of hot-air-dried chips. The combination of RF-assisted air drying and partial recirculating of dryer exhaust air achieved the highest saving in TEC of about 12.4%, along with a lower moisture absorption capacity and no deterioration of product quality attributes. This drying concept is therefore recommended for the industrial drying of several food materials.

## 1. Introduction

*Momordica charantia* Linn., also known as bitter melon, is a functional vegetable that is associated with various health benefits to consumers because it possesses several bioactive substances, including antioxidants, alkaloids, and polypeptides [1]. The fruit pulp of bitter melon is rich in antioxidants, including ascorbic acid, gallic acid, phenolic, saponins, flavonoid, and chlorophyll [2]. The content of these beneficial compounds depends on the processing conditions. Fresh bitter melon has a high water content and often contains a certain load of microorganisms, which requires proper preservation to minimize biological deterioration and retain the maximum nutritional value of the original plant. Despite the high nutritional value of bitter melon, it is not a popular vegetable among consumers because of its bitter taste. Some novel alternate products containing bitter melon are needed which mask the bitterness to some extent. In this study, we propose blending bitter melon with apple puree to make a restructured product that is palatable to the consumer. Apple is one of most consumed and healthy fruits around the world; it is also rich in bioactive compounds such as phenolics and vitamin C [3]. Restructured foods are made of a mixture of different ingredients, enabling the design of products with more desirable properties from their original components [4]. Hence, this study was targeted towards the development of restructured apple and bitter melon chips to minimize the bitter taste, and provide a new healthy product to consumers.

Thermal drying results in reduced water activity to levels that prevent microbial growth. It offers numerous benefits in extending the shelf life of dried products; these include simplified handling and storage, reduced packaging, and lower transportation costs [5]. Therefore, this method is implemented widely for the drying of food products including vegetables, sea food, grains, snacks, spices, and other foodstuffs [6,7]. Convective hot-air drying (HAD) is an intensively utilized technique for commercial production of numerous foodstuffs [5]. It is estimated that more than 80% of industrial dryers are convective HAD systems. HAD normally requires longer drying duration and higher energy input [8]. Consequently, this method may result in final products with an undesirable color, low rehydration capacity, aroma degradation, severe shrinkage, as well as substantial nutraceutical component deterioration [8,9]. In recent years, various emerging drying technologies were considered and commercialized to enable the introduction of energy-saving approaches without compromising on the quality of the finished products. Several novel techniques including dielectric drying [10], infrared-freeze drying [11], low-pressure superheated steam drying [12], as well as impingement jet drying [13] have been implemented for the processing of foodstuffs.

The electromagnetically generated dielectric energy (microwave and radio frequency) is a result of intensive friction between the polar water molecules and ions due to the rapid change of the electromagnetic field [14]. Various studies have confirmed that microwave (MW) drying has many advantages for food drying, such as a rapid drying rate [7], short processing cycle, rapid and accurate control, as well as energy saving [14]. However, MW drying also has several shortcomings, which include non-uniform drying due to uneven electromagnetic field distribution, potential structural damage, the high degradation of valuable ingredients, and the short penetration depth of microwaves into the material [7,14]. Compared to two commonly used MW frequencies (915 MHz and 2450 MHz), RF has longer wavelengths that enable RF energy to penetrate further through the absorbing material and provide better heating uniformity [15]. Hence, RF drying is a promising technique which can provide dried products with high quality due to fast and volumetrically electromagnetic heating patterns, deep energy penetration, and a more reliable control of product temperature [15,16].

Drying is the most energy-intensive unit operation and accounts for 20~25% of the total energy needed in the food industry [17]. The food industry needs to meet the demands for reducing the cost of energy as well as decreasing CO_2_ (carbon dioxide) emissions. RF drying on its own has the drawback of a low drying efficiency at the earlier stage, resulting in a relatively slow drying rate [18]. To overcome this disadvantage, the RF-assisted hot-air drying (RFHAD) technique is developed to improve drying performance whilst maintaining the high quality of dried foodstuffs.

Although RFHAD has already been applied to stem lettuce, carrot, and tilapia [19,20,21], its effect on the drying performance and key quality attributes of restructured bitter melon and apple chips has not been studied yet. Moreover, exhaust air recirculation (EAR) is commonly integrated into HAD to further achieve a high energy efficiency, but EAR is not yet applied to the RFHAD processing of food products. Therefore, the target of this research was to verify a new energy saving approach based on RFHAD with EAR for producing restructured bitter melon and apple chips. The quality of the restructured chips was evaluated by means of color, moisture absorption characteristics, bioactive compound content, and other attributes. Meanwhile, the drying characteristics, average drying rate, total energy consumption, and effective moisture diffusivity were also compared.

## 2. Materials and Methods

### 2.1. Materials and Sample Preparation

The fully matured apples (*Fuji*) and fresh bitter melons (*Momordica charantia* Linn.) of 80~90% maturity (maturity determined by the appearance, weight, and seed color of the bitter melons) utilized in this investigation were obtained from a local grocery store (Vanguard, Wuxi, China). Before each series of experiments, the fruits were peeled after washing, and then the stems and seeds were removed.

As shown in Figure 1, the restructured bitter melon and apple chips were made according to the following procedure:The washed and treated bitter melon and apple were milled for 3 min using a miller (model: L18-Y915S, Joyoung, Hangzhou, China) until the puree became smooth.The puree was weighed and then mixed with white sugar, sunflower oil, and corn starch to form the dough. The formulation (100 g) of the mixture was as follows: bitter melon puree 28 g, apple puree 20 g, corn starch 46 g, white sugar 4 g, and sunflower oil 2 g.After manually malaxating the dough for 5 min, the dough was evenly pressed to a thickness of 5 mm, and then the circular chips were formed using a baking mold with a diameter of 10 mm.
Figure 1Schematic diagram of restructured bitter melon and apple chips produced using different drying methods.
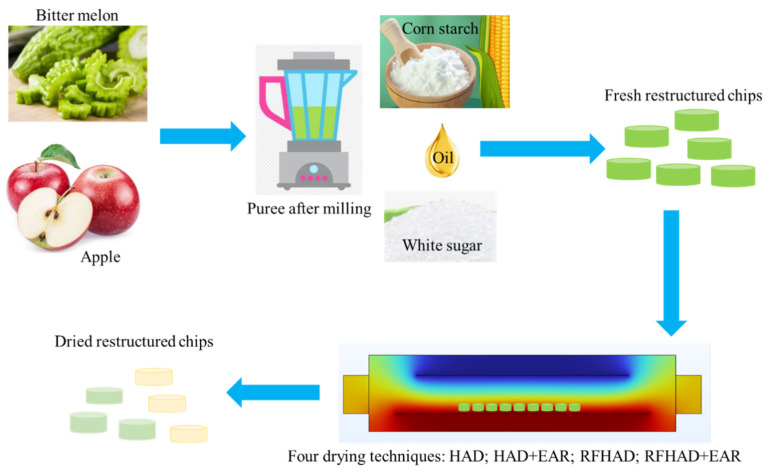



### 2.2. Drying Equipment

A lab-scale RF heating device (RF power:6-kW; frequency: 27.12 MHz; type: SO-6B, Monga Strayfield, Pune, India) integrated with a convection air heating system was employed in the drying experimental trials. Figure 2 depicts the basic diagram of this combined drying instrument. The RF generator consists of a moveable top flat electrode and a static bottom flat electrode; this generator is used for generating heat inside samples in the RF cavity. The variable electrode gap between two electrodes (100 mm to 400 mm) is used to control the intensity of electromagnetic energy during RF heating. Hot air is additionally supplied through a stainless-steel soft tube which is connected to an electrical heater. The air temperature is adjustable from ambient temperature to 110 °C with the air speed up to 3 m/s. A fan is used for removing the exhaust air from the RF chamber, and a part of the dryer exhaust air can be recycled into the dryer chamber using a three-way valve.

### 2.3. Experiment Procedures

The restructured bitter melon and apple chips (300 ± 0.3 g per batch) were put in a Teflon tray (dimension: L = 300 mm; W = 200 mm; H = 50 mm), and the Teflon tray was loaded on the middle part of the bottom electrode inside the heating chamber. The samples were subjected to HAD and RFHAD with/without EAR. Process parameters and quality indicators (including drying time, moisture content, color, bioactive compounds, moisture sorption isotherms, and so on) were measured during/after drying. The experimental matrix of this study is presented in Table 1.
foods-13-00197-t001_Table 1Table 1Experimental matrix implemented in this study.Drying MethodsMeasurementsHADDrying time, total energy consumption, effective moisture diffusivity, color, textual property, volume and dimension changes, content of chlorophyll, vitamin C content, and total phenolic content, antioxidant capacity, pH and titratable acidity, enzyme activity, moisture content and moisture absorption isothermsHAD + EARRFHADRFHAD + EARHAD, hot-air drying; HAD + EAR, hot-air drying with exhaust air recirculation; RFHAD, radio-frequency-assisted hot-air drying; RFHAD + EAR, radio-frequency-assisted hot-air drying with exhaust air recirculation.


For HAD and HAD + EAR, the air temperature and speed were fixed at 60 °C and 1.5 m/s, which could obtain a relative faster drying rate and avoid excessive energy consumption. According to the preliminary tests, EAR increased the relative humidity (RH) of hot air which decreased the drying rate, especially at the early stage of the drying process. Hence, EAR was only implemented after half of the required drying time had elapsed, in order to avoid the negative impact of hot air with high humidity.For RFHAD and RFHAD + EAR, the electrode distance was set at 150 mm which yielded a faster heating rate without arc discharge. The air temperature and speed were also fixed at 60 °C and 1.5 m/s, respectively. EAR was only implemented until half of the drying process was completed.

During drying, chips were withdrawn at regular intervals (every 30 min and every 15 min near the end of drying) to determine the moisture content (MC) and other quality indicators. The procedure was repeated until the MC of dried chips was less than the targeted value of 0.10 g/g (d.m.). The dried chips were properly stored in aluminum foil films for other measurements.

### 2.4. Moisture Content

An oven method was applied to examine the MC level of samples according to China National Standard (GB 5009.3-2016 [22]). Restructured chips were withdrawn at regular intervals during processing and placed in an oven at 105 °C until a constant weight was obtained. Drying kinetics were described based on the MC variation at a given time and the MC was calculated by using the following formula:(1)$$X_{t}=\frac{m_{t}-m_{f}}{m_{f}}$$
In Equation (1), X_t_ represents the MC at time t on a dry matter (g/g d.m.), m_t_ represents the product weight at time t (g), and m_f_ represents the weight of the completely dried substance (g).

### 2.5. Drying Rate

The drying rate (DR) represents the speed of the drying process and was calculated based on the following formula:(2)$$DR=\frac{m_{t+\Delta t}-m_{t}}{m_{f}\cdot \Delta t}$$
where m_t+Δt_ represents the product weight at time t+Δt (g), and Δt represents the time interval of drying (h).

### 2.6. Effective Moisture Diffusivity

The drying process and the transport of moisture inside biological materials were described using Fick’s second law of diffusion. A uniform moisture distribution, a negligible external resistance, a constant effective moisture diffusivity (D_eff_), and a negligible shrinkage of samples were assumed during the drying process, and the moisture ratio change (MR) was calculated using the following formula:(3)$$MR=\frac{8}{\pi^{2}}\sum_{n=1}^{\infty }\frac{1}{(2n-1{)}^{2}}\exp(-{\left(2n-1\right)}^{2}\pi^{2}t\frac{D_{eff}}{4L^{2}})$$
where MR represents the moisture ratio change within a given time, L represents the half-thickness of dried samples (m), t represents the drying duration (s), and n is a positive integer.

The MR of the restructured chips was computed as follows:(4)$$MR=\frac{X_{t}-X_{e}}{X_{0}-X_{e}}$$
where X_0_, X_e_ represent the MC (g/g, d.m.) of fresh and dried restructured chips.

For a sufficiently long drying duration, all terms except the first within the above series were negligible. Thus, Equation (3) can be simplified as Equation (5):(5)$$MR=\frac{8}{\pi^{2}}\exp(-\pi^{2}t\frac{D_{eff}}{4L^{2}})$$

Further, Equation (5) can be changed into a logarithmic form as Equation (6):(6)$$Ln(MR)=Ln(\frac{8}{\pi^{2}})-(-\pi^{2}t\frac{D_{eff}}{4L^{2}})$$

From Equation (6), a plot of Ln(MR) versus the drying duration gave a straight line with the slope of k. D_eff_ can be calculated from Equation (7):(7)$$D_{eff}=\frac{4L^{2}Slope\left(k\right)}{\pi^{2}}$$

### 2.7. Energy Consumption

A watt-hour meter (model: DTS-634, Zhejiang CHINT Electrics Co., Ltd., Wenzhou, China) was installed to determine the total energy consumption (TEC) during the processes.

The specific energy consumption (SEC) can be determined using Equation (8):(8)$$SEC=\frac{{TEC}_{e}}{m_{l}}$$
where TEC_e_ and m_l_ are the TEC for each drying process (kW·h) and the weight loss of samples during drying (kg), respectively.

### 2.8. Color Measurements

The color measurements were conducted on raw and dried samples using a colorimeter under conditions of the standard illuminant D65 and 10 degrees observer (model: BC-10, Minolta, Osaka, Japan). The color values were expressed according to the International Commission on Illumination (CIE) color coordinates L*, a*, and b* of tested samples. The total color difference (ΔE) was calculated from the color coordinates using the following formula:(9)$$\Delta E=\sqrt{(L_{0}-L*{)}^{2}+(a_{0}-a*{)}^{2}+(b_{0}-b*{)}^{2}}$$
where L_0_, a_0_, and b_0_ represent the color value of the raw chips.

In addition, chroma (C) and hue angle (H) were obtained using the following expressions:(10)$$C=\sqrt{(a*{)}^{2}+(b*{)}^{2}}$$
(11)$$H=\arctan(\frac{b*}{a*})$$

### 2.9. Chlorophyll, Vitamin C, Total Phenolic, and Total Flavonoid Content

The methodology introduced by Potisate et al. [23] was applied to examine the chlorophyll content. Dried restructured chips (3 g) were weighed, and ground with 0.1 g CaCO_3_ and 80% acetone in a mortar and pestle. After the ground powder was fully disintegrated, the mixture suspension was filtered and the filtrate was washed using 80% acetone at least three times. The filtrate was put into a volumetric flask and diluted with 80% acetone to 50 mL. The filtrate absorbance of the chlorophyll extract solution was examined using a UV-visible spectrophotometer against a blank under the wavelengths of 663 and 645 nm. The total chlorophyll content of the restructured chips was calculated as follows:(12)$$Chlorophyll\;content\;(mg/100\;g)=\left(8.05A_{663}+20.29A_{645}\right)\times \frac{v}{m}\times 100$$
where A_663_ represents the absorption value measured under the wavelength of 663 nm, A_645_ represents the absorption value measured under the wavelength of 645 nm, v is the volume of chlorophyll extract (mL), and m is the weight of dried product (g).

As described by Rahman et al. [24], the quantitative discoloration of 2,6-dichloro-indophenol method was used and modified for examining the vitamin C (Vc) content. Dried chips (3 g) were fully ground and dissolved with 20 mL extraction reagent. The mixture suspension was filtrated and later, 10 mL of filtrated solution was titrated with calibrated 2,6-dichloro-indophenol solution for 15 s until the color changed to pink. The Vc content was expressed in micrograms per 100 g dry matter (mg/100g d.m.).

With a few alterations, the Folin–Ciocalteu colorimetric method was applied for measuring the total phenolic content (TPC) of the samples [25]. Dehydrated chips (3 g) were fully ground and blended with 1.25 mL of Folin–Ciocalteu agent and 10 mL ultrapure water. The extract solution was stored at 4 °C for 6 min and then it was filtered. A total of 1.0 mL of filtrated solution was further mixed with 12.5 mL 7.5% Na_2_CO_3_, and afterwards the solution was kept in complete darkness for 40 min under room temperature. Finally, the absorbance of total phenolic extract was examined in the UV-visible spectrophotometer at the wavelength of 760 nm. The concentration of total phenolic compounds was presented as milligram gallic acid equivalent (GAE) per gram of dry matter product (mg GAE/g d.m.).

Three grams of fully ground chips, 12 mL ultrapure water, and 1.80 mL sodium nitrite (75 g/L) were mixed and kept for 10 min. Afterward, 9.0 mL aluminum chloride solution (10 g/L) was dosed and kept for 10 min, and 4 mL sodium hydroxide solution (40 g/L) was added in sequence. Then, the mixture suspension was filtrated and kept for 35 min. Subsequently, the filtrate absorbance of total flavonoid content (TFC) extract was determined using a UV-visible spectrophotometer against a blank at the wavelength of 510 nm. The calibration curve was formed based on the standard rutin solution, and the experimental data were presented as milligrams of rutin equivalents per gram dry matter (mg·RE/g d.m.).

### 2.10. Antioxidant Capacity

The extracts of restructured chips were applied to examine the antioxidant capacities including 2,2-diphenyl-1-picrylhydrazyl (DPPH) scavenging, 2,2′-azino-bis 3-ethylbenzothiazoline-6-sulfonic acid (ABTS) radical scavenging, as well as ferric-reducing antioxidant power (FRAP). The extraction method introduced by Qiu et al. [26] was used with minor alterations. Ground restructured chips were weighted (3 g) and evenly dissolved with 150 mL of 95% ethanol; then, the mixture was extracted for 60 min under an ultrasonic condition (frequency: 30 kHz, power: 300 W). The supernatants were collected from the mixture solution after 12 h for further analysis.

With a few alterations, the method introduced by Qiu et al. [26] was applied to measure the DPPH of the restructured chips. Generally, an amount of 0.3 mL of the extract or distilled water (as reference) was dissolved with 14.7 mL DPPH solution (20 mg/L), and then kept away from light for 45 min. The absorbance of DPPH extract solution was examined using a UV-visible spectrophotometer at the wavelength of 517 nm. Trolox solution was used to conduct the calibration curve and the DPPH result was expressed as μmol of Trolox equivalents per gram dry matter (μmol TE/g d.m.).

The method introduced by Zhang et al. [27] was applied for performing the determination of ABTS. Briefly, the ABTS radical cation (ABTS+) was generated by mixing an aqueous ABTS solution (3.84 g/L) with potassium persulfate solution (0.66 g/L) away from light for 16 h. Further, the mixture was diluted with phosphate-buffered solution (pH 7.4) to an absorbance at 734 nm of 0.70 ± 0.01. Afterwards, the 0.3 mL extract was added to 29.7 mL of ABTS + solution or distilled water (as reference) and stored in complete darkness for 20 min. The absorbance of the ABTS extract solution was checked using the UV-visible spectrophotometer at the wavelength of 734 nm. Trolox solution was also applied to conduct the calibration curve and the final result was expressed as μmol TE/g d.m.

The FRAP activity was examined according to the description from Qiu et al. [26] with slight alterations. Generally, the FRAP agent consisted of FeCl_3_ (20 mM), TPTZ (10 mM), and acetate buffer (0.1 M, pH 3.6), with a volume ratio of 1:1:10. Afterwards, 0.3 mL supernatant was pipetted to 7.2 mL of FRAP agent and the mixture was kept away from light for 20 min. The absorbance was then measured using the UV-visible spectrophotometer at the wavelength of 593 nm. Trolox solution was used as the standard and distilled water as the blank control. The concentration of FRAP was presented as μmol TE/g d.m.

### 2.11. pH and Titratable Acidity

Fully ground chips (3 g) were diluted with 30 mL ultrapure water for 30 min. The mixture solution was filtered and collected for further analysis.

The pH of solution was examined by a pH meter (type:3100, Ohausyt, Shanghai, China), which was calibrated before measurement according to the AOAC method.

The titratable acidity of the restructured chips was examined using the phenolphthalein method introduced by Kahraman et al. [28] with minor alterations. The mixture solution was combined with phenolphthalein as indicator, and afterwards the filtrated solution was titrated with 0.1 N NaOH solution until pH 8.1 was reached. The titratable acidity was presented as mg citric acid/100 g (d.m.).

### 2.12. Enzyme Activity

Fully ground chips (3 g) were diluted with extraction buffer solution (60 mL) in complete darkness for 60 min at 4 °C, and then centrifuged under 12,000× *g* for 25 min at 4 °C. Then, the mixture suspension was filtered and collected for further analysis.

Polyphenol oxidase (PPO) enzyme activity was tested using the method introduced by González et al. [29] with slight alterations. The reaction mixture contained 0.5 mL catechol (50 mmol/L) and 2 mL sodium phosphate buffer. Then, the absorbance of the mixture was determined every 15 s for 120 s after adding 1.0 mL filtrated solution, using a spectrophotometer at the wavelength of 420 nm. One unit of PPO results was expressed as 0.01 unit change of absorbance unit per min (U·min^−1^g^−1^).

The filtrated solution was used to examine the peroxidase (POD) activity according to the method described by Xing et al. [30]. The reaction mixture consisted of 1.0 mL filtrated suspension and 3.0 mL guaiacol solution (25 mmol/L). Then, the absorbance of mixture was determined every 15 s for 2 min after dosing 0.2 mL H_2_O_2_ solution (0.5 mol/L), using a spectrophotometer at the wavelength of 470 nm. One unit of POD results was expressed as 0.01 unit increase of absorbance unit per min (U·min^−1^g^−1^).

### 2.13. Texture

The hardness of the dehydrated restructured chips was examined by a texture analyzer (type: TA-XT2i, Stable Micro Systems Ltd., Vienna Court, Surrey, UK). Samples were placed on the table of the texture analyzer; the samples were penetrated with the stainless-steel cylindrical penetrometer probe (P5: diameter of 5 mm). The following setting steps and parameters were performed [31]: a pre-speed of 1 mm/s, a test speed of 0.5 mm/s, a post-speed of 10 mm/s, a trigger force of 5 g, and a puncture distance of 15 mm. The Texture Exponent software (version: 32) was used for computing the force–time curve, and the peak force of the first compression cycle on the force–time curve was taken as the hardness.

### 2.14. Volume and Dimension Changes

Volume changes due to shrinkage were measured using the displacement method reported by Zhang et al. [32] with minor modifications. The replacement material was sea sand after screening with a sifter with a size of 0.4 mm. Ten chips were measured and the mean values are reported. The shrinkage ratio was calculated using the following formula:(13)$$SR=\frac{V_{i}-V_{f}}{V_{i}}\times 100$$
where SR is the shrinkage ratio (%), V_i_ represents the initial volume of the fresh products (cm^3^), and V_f_ represents the final volume of the dried chips (cm^3^).

Wang et al.’s [33] vernier caliper method with minor modifications was employed to determine the dimensional changes of the dried chips. The thickness and diameter of the restructured chips were determined at five different positions; eight samples were randomly taken from each batch and the average value was reported. The diameter change (Dc) and thickness change (Tc) were calculated using the following formulas:(14)$$D_{C}=\frac{D_{0}-D}{D_{0}}\times 100$$
(15)$$T_{C}=\frac{T_{0}-T}{T_{0}}\times 100$$
where D and T are the diameter and thickness of the processed products (mm), and D_0_ and T_0_ are the diameter and thickness of the fresh products (mm).

### 2.15. Water Activity and Moisture Adsorption Behavior

A water activity meter (model: 4TEV, Decagon Devices, Pullman, WA, USA) was used to examine the water activity of the dehydrated chips. The device was calibrated before determinations and samples were measured at 25 °C.

The moisture sorption isotherms of the dehydrated products were quantified using the ProUmid SPS instrument (model: x-1, ProUmid GmbH, Ulm, Germany). This device enables the automatic gravimetric measurement of the moisture sorption of multiple products with controlled RH and temperature. It also enables the weighing of the mass of the samples with high accuracy (0.001 mg) and its recording using a SPSx-1 moisture sorption analyzer during the measurements. Dried restructured samples were placed on the sample pans and measured at equilibrium between 15~75% RH and 25 ± 0.1 °C. The determination cycle was started at 15% RH and ended at 75% RH with steps of 15%. The determination time was set at 24 h. The sample weight change ratio (dm) was computed using SPS-Toolbox software (version: 1.15).

A typical Guggenheim–Anderson–de Boer (GAB) model was chosen to describe the moisture sorption isotherms of the dehydrated chips. The GAB model was determined mathematically using the following equation:(16)$$M_{eq}=\frac{M_{m}CKa_{w}}{\left(1-Ka_{w})(1-Ka_{w}+CKa_{w}\right)}$$
where $M_{eq}$ denotes the equilibrium MC (g/g d.m.), $M_{m}$ denotes the monolayer MC (g/g d.m.), $a_{w}$ denotes the water activity, and C and K denote the constants.

Further, the GAB equation (16) can be transformed into a second-order polynomial Equation (17) to easily calculate the parameters ($M_{m}$, C, and K) of the GAB model [34]:(17)$$\frac{a_{w}}{M_{m}}=Aa_{w}^{2}+{Ba}_{w}+D$$

Further, K is the solution of the following quadratic equation:$${AK}^{2}+BK+D=0$$

Further, $M_{m}$ and C can be calculated using the following equations:$$M_{m}=\frac{1}{B+2AK}$$
$$C=\frac{B}{AK}+2$$

### 2.16. Principal Component Analysis (PCA) and Correlation Analysis (CA)

PCA and CA were performed via the use of OriginPro software (version: 2023b, Origin Lab, Northampton, MA, USA) to identify specific correlation patterns of all interested parameters among the restructured chips produced using different drying techniques. The average values of color difference, chlorophyll content, TPC, TFC, Vc content, antioxidant capacity, pH, titratable acidity, enzyme activity, hardness, volume change, dimension change, MC, water activity, as well as the average levels of drying duration, D_eff_, TEC, and SEC were also involved in the PCA and correlation plot analysis.

### 2.17. Statistical Analysis

Unless otherwise stated, the results of the tests are based on three replicates and expressed as average ± SD. The data analysis was performed using Duncan’s multiple-range analysis of variance (ANOVA) using the SPSS statistics software (Version 25.0, SPSS: Armonk, NY, USA). For the experimental results, a statistical difference was considered at a confidence interval of 0.95 (*p* < 0.05).

## 3. Results and Discussion

### 3.1. Drying Characteristics

The initial MC of the raw restructured products was 0.488 ± 0.02 g/g d.m., and the products were dehydrated to the MC below 0.10 g/g d.m. using different drying techniques. The drying curves drafted from experimental MC values (d.m.) are presented in Figure 3a. HAD + EAR took the longest time, with a drying duration of 7.5 h, which decreased to 6.1 h with HAD, 4.2 h with RFHAD + EAR, and 3.7 h with RFHAD. It was found that about 38.8% reduction in drying duration was obtained using the RFHAD method compared to the HAD method, and the drying duration of RFHAD + EAR was reduced by about 30.5% when compared to that of HAD + EAR. This can be explained by the fact that the electromagnetic energy used in RF drying generates volumetric heat, which can clearly increase the drying rate in contrast to surface heating in convective drying [16]. Meanwhile, RF energy heats inside and causes moisture to evaporate to the surface, and then hot air facilitates moisture migration when samples are processed by RFHAD with/without EAR. Thus, the combination of hot air and RF energy provides a novel approach to improve the heat and mass transfer performance of HAD. According to the results from Roknul et al. [19], the RF drying of stem lettuce had a notable reduction in drying duration of up to 66.7%.

The curve of the drying rate versus MC (d.m.) is shown in Figure 3b. It was observed that RFHAD and RFHAD + EAR had a relative higher drying rate compared to HAD and HAD + EAR in the initial stage of the dehydration process. This could be because RF radiation energy induced heat generation within the matrices through molecular friction caused by the oscillation of molecules and ions, which resulted in an enhanced drying rate. Similar findings were announced for the RFHAD production of carrot and orange peel [20,35]. It was also found that the drying rates of the four different drying techniques were quite close during the later stage, and that the drying rates gradually decreased with decreasing MC. Both HAD and RFHAD + EAR showed a relative slower drying rate compared to that of HAD and RFHAD due to the fact that EAR was employed during the second half of the drying duration. This may be ascribed to the implementation of EAR, which brings more moisture in hot air, which hindered the removal of water from the surface of the dried samples during the drying process [36].

### 3.2. Effective Moisture Diffusivity

The D_eff_ value was calculated from the experimental results, which represented the overall mass and moisture transfer property of the dried samples [37]. Figure 4 shows the different drying techniques’ variation in D_eff_ values, which were between 3.218 × 10^−9^ and 6.889 × 10^−9^ m^2^/s; these results were in the general range from 10^−12^ to 10^−6^ m^2^/s for the drying of biomaterials [38]. As expected, the higher D_eff_ values were obtained for RFHAD (6.889 × 10^−9^ m^2^/s) and RFHAD + EAR (6.062 × 10^−9^ m^2^/s), whereas the lower D_eff_ values were obtained for HAD (4.124 × 10^−9^ m^2^/s) and HAD + EAR (3.218 × 10^−9^ m^2^/s). The D_eff_ value of HAD at 60 °C was consistent with previous publications [39,40], ranging from 3.210 × 10^−9^ to 2.840 × 10^−8^ m^2^/s for the drying of tomato and P. macrocarpa fruits with similar geometries. Furthermore, the D_eff_ value of the RFHAD method were 1.67 times higher compared to that of HAD, which demonstrated that the moisture transfer efficiency of the RFHAD process was much higher than that of HAD. Similar to this finding, Elik et al. [41] reported that the D_eff_ value of RFHAD was nearly 1.51 times higher than that of HAD for the drying of black carrot. This result also further proved that the volumetric heating pattern of RFHAD facilitated the diffusion and transfer of moisture, thus achieving a larger D_eff_ value. In contrast, the direction of mass and heat transfer was opposite during HAD, which impeded the moisture transfer and resulted in a smaller D_eff_ value.

### 3.3. Energy Aspects

Energy consumption is a key aspect in evaluating the effectiveness of the proposed drying methods for the dehydration of restructured chips. The TEC values of different drying techniques are depicted in Figure 5. The highest TEC value for the drying of restructured chips was observed in HAD + EAR (31.87 kW·h), followed by HAD (28.59 kW·h) and RFHAD (27.51 kW·h), whereas RFHAD + EAR consumed the least, with a total energy consumption of 25.05 kW·h. Although experimental conditions with limited amount of samples were applied to obtain a better heating uniformity, the TEC value of RFHAD was still reduced by 3.8% compared to that of HAD. These results are related to the reduction in drying time by applying RF energy during HAD, where this enabled a high percentage reduction in drying duration. Similar results on energy saving in the RFHAD of hazelnuts have been published by Wang et al. [42]. Meanwhile, it was clearly seen that the use of EAR on RFHAD contributed a significant (*p* < 0.05) energy reduction of 8.9% and 12.4% compared to that of RFHAD and HAD, respectively. It was proved that the reuse of exhaust air is a promising energy saving approach that could further reduce the energy consumption of the drying process. Müller and Heindl [43] also reported that the use of exhaust air combined with ambient air could contribute a reduction of 44~85% in energy consumption for the drying of two medical plants. However, the drying time of HAD + EAR was prolonged by 23.0% in comparison with that of HAD, resulting in a higher increase in total energy consumption. This could be attributed to the use of EAR which increases the air relative humidity during the drying process; this limits the mass transfer across the air and sample surface, which leads to an increase in the processing duration as well as energy consumption [44,45].

The SEC values were calculated from the experimental drying data, which represented the required energy to remove moisture during the drying process [46]. The highest SEC values were observed in HAD (348.42 kW·h/kg) and HAD + EAR (396.18 kW·h/kg), whereas the lowest SEC values were achieved in RFHAD (328.79 kW·h/kg) and RFHAD + EAR (301.57 kW·h/kg). This can be explained by the fact that RF energy generates volumetric heat which can largely increase the drying rate in comparison to low heat and mass transfer efficiency during HAD. In the case of the RFHAD + EAR method, the SEC value was reduced by 8.3% in contrast to that of RFHAD. This result demonstrated that energy efficiency was improved by implementing the air recirculation approach during RFHAD. Additionally, the SEC values of different drying techniques closely followed the drying duration. Shewale et al. [46] also found similar results when RF energy was used for the processing of apple slices.

### 3.4. Color

Color is the key factor in consumers’ judgment of food quality, and is therefore a major consideration of product quality related to consumer acceptance [47]. Drying methods and their parameters (such as temperature, drying time, and oxygen) have a huge influence on the color of the final products. The color parameters of chips produced using HAD and RFHAD with/without EAR are shown in Table 2. Fresh restructured bitter melon and apple chips exhibited the highest brightness (L* = 70.17 ± 0.43), the highest greenness (a* = −5.34 ± 0.28), and the lowest yellowness (b* = 16.97 ± 0.32). Compared to fresh samples, a significant decrease of “L*” values was found for all the dehydrated chips (*p* < 0.05); the reduction in the L* value indicated the darkening of the dried chips. The coordinate “a*” values of RFHAD and RFHAD + EAR chips were significantly smaller compared to chips produced using HAD and HAD + EAR (*p* < 0.05); this might be explained by the fact that more chlorophyll was degraded under the longer drying duration of those drying processes. Meanwhile, the coordinate “b*” values of RFHAD + EAR and RFHAD chips were significantly smaller than those of HAD and HAD + EAR chips. This implied that the restructured chips dried using HAD and HAD + EAR presented more yellowish pixels compared to chips dried using RFHAD and RFHAD + EAR. This could be attributed to the fact that a longer drying cycle exacerbated the Maillard reactions, caramelization reactions, and the ascorbic acid browning that occurred during the drying process. A similar phenomenon was reported in the drying of lotus pollen [48]. Furthermore, samples dried using HAD + EAR had the biggest color difference (ΔE = 17.37), followed by HAD (ΔE = 14.87), RFHAD + EAR (ΔE = 6.52), and RFHAD (ΔE = 5.41). HAD + EAR and HAD products indicated notably higher ΔE values than those of the RFHAD + EAR and RFHAD samples (*p* < 0.05). The trend of ΔE from different drying techniques had a positive correlation with the drying time; this could be explained by the fact that a prolonged drying time with the presence of oxygen causes color changes in the final products during the thermal dehydration process. The same finding was stated by Xie et al. [49], who observed that RF-assisted HAD samples had a better color compared to that of HAD. Furthermore, the restructured chips produced using the RFHAD + EAR and RFHAD methods displayed a negligible difference in color difference (*p* < 0.05).

As seen from Table 2, C values increased for all drying techniques, and a very close correlation was observed between C and b* values. The results presented an increase in C values along with the increase in drying time. Meanwhile, the results also show a significant decrease in H values for all dried chips compared to the fresh restructured chips (*p* < 0.05). A similar result was published by Mahmood et al. during the RF-assisted hot-air drying of paddy [50]. Furthermore, fresh samples with a H value of about 107° suggested that the fresh samples were within the region of yellow-green (90° < H < 180°). As can be observed in Table 2, the RFHAD and RFHAD + EAR samples were within the same region as the fresh samples. However, the H values of chips dried using HAD and HAD + EAR were significantly decreased (*p* < 0.05), which indicated that the color of the restructured chips shifted from yellow-green to yellow-red (0° < H < 90°). Overall, both the C and H values of the RFHAD/RFHAD + EAR samples were closer to those of the fresh restructured chips.

### 3.5. Chlorophyll, Vitamin C, Total Phenolic, and Total Flavonoid Content

As shown in Table 3, a significant difference in chlorophyll content was found between RFHAD/RFHAD + EAR and HAD/HAD + EAR processed chips (*p* < 0.05). HAD and HAD + EAR processed chips showed the lowest chlorophyll retention due to their prolonged drying cycles, resulting in a severe thermal damage of chlorophyll. Shewale et al. [51] also found a higher chlorophyll retention for *Rosmarinus officinalis* dried using RFHAD than that of HAD samples. Moreover, samples dried using RFHAD had a higher chlorophyll retention (28.17 mg/100 g), which was aligned with the observation of the lowest a* value (better greenness) among all drying methods in Table 2.

Vc is an oxygen and heat-sensitive nutrient and susceptible to degradation during the heating process [52]. It can be noticed in Table 3 that the RFHAD + EAR samples had the highest Vc content of 8.58 mg/100 g, followed by RFHAD, HAD, and HAD + EAR samples, with 8.37, 5.07, and 4.22 mg/100 g, respectively. A notable difference was observed between RFHAD/RFHAD + EAR and HAD/HAD + EAR chips (*p* < 0.05), because the longer drying duration required by HAD led to a severe thermal degradation of Vc. Similar trends, with RFHAD enhancing the retention of Vc content compared to HAD were observed in the processing of carrot, bitter melon, and mango [16,32,53]. Moreover, the results also revealed that there were no statistically differences among samples dried using the RFHAD and RFHAD + EAR methods (*p* < 0.05).

Phenolic compounds are bioactive substances occurring widely in foodstuffs and normally acting as natural antioxidants [54]. The TPC values of the four drying processes are shown in Table 3. As expected, significantly higher TPC values were obtained from samples dried using RFHAD and RFHAD + EAR due to their shorter drying duration compared to those of HAD and HAD + EAR (*p* < 0.05). Similar results were also obtained in the drying of hazelnuts, strawberry, and black carrot pomace using RF energy [41,42,55]. Meanwhile, it was observed that there was no statistical difference among samples produced using RFHAD and RFHAD + EAR (*p* < 0.05).

The TFC values of the restructured chips are also listed in Table 3. The lowest retention of TFC was observed in HAD + EAR (2.32 mg·RE/g) samples, followed by HAD samples, with a TFC value of 2.65 mg·RE/g. This was because a longer drying duration led to more exposure of the flavonoid compounds to drying effects. Chips produced using RFHAD and RFHAD + EAR showed a noticeable (*p* < 0.05) increase in TFC retention of 38.5% and 17.0% compared to that of HAD samples. Despite the shorter drying duration of those two dielectric drying methods, the appropriate dielectric energy might cause the breakdown of cellular constituents that might induce the release of more flavonoids from the samples [56]. A similar impact of the drying process on TFC was reflected in previous publications regarding the RF drying of purple-fleshed potato and Sichuan pepper [57,58].

### 3.6. Antioxidant Capacity

The antioxidant capacities of the restructured chips were evaluated in terms of DPPH, ABTS, and FRAP assays. As presented in Table 4, the experimental results showed that the four implemented techniques had a significant impact on the DPPH, ABTS, and FRAP values. The highest antioxidant activities were found in RFHAD chips, with statistically differences in ABTS and FRAP assays (*p* < 0.05). More specifically, it was observed that RFHAD samples obtained the highest DPPH (9.46 μmol TE/g), while the HAD + EAR treatment showed the lowest DPPH (6.33 μmol TE/g). A similar fact was also observed by Ai et al. [59], who also highlighted that RFHAD achieved the highest DPPH scavenging ability in comparison with that of the solo HAD and solo RF drying of Amomi fructus. The ABTS values of RFHAD (58.11 μmol TE/g) and RFHAD + EAR (54.27 μmol TE/g) samples were higher compared to those of HAD (44.97 μmol TE/g) and HAD + EAR (42.10 μmol TE/g) samples, and decreased along with an increase in drying duration. Similarity, the FRAP results had a similar trend as observed in the determination of DPPH and ABTS results. This might mean that the RF process could maintain more antioxidant components due to less thermal and chemical degradation, as it was conducted within a shorter drying duration. These findings were in accordance with a previous study on the RF drying of purple potato and dandelion leaves [57,60]. Moreover, HAD + EAR chips obtained a significantly lower FRAP level in contrast to that of HAD (*p* < 0.05). Wojdyło et al. [61] reported that the FRAP values were affected by antioxidant components, which were easily thermally degradable due to a prolonged drying duration at 60 °C.

In addition, the antioxidant activities of dehydrated products may be associated with the content of bioactive components, because these components act as scavengers of free radicals during the oxidation reaction [62]. The higher antioxidant capacities of RF-produced chips were supported by the higher retention of chlorophyll, Vc, TPC, and TFC in chips dried using RFHAD and RFHAD + EAR. Many studies proved that the higher retention of bioactive components in RFHAD products led to enhanced antioxidant activities [51,59].

### 3.7. pH and Titratable Acidity

Different drying techniques showed variable impacts on the pH values of dehydrated chips. As noted in Figure 6, the chips treated using RFHAD and RFHAD + EAR reached a lower pH range from 4.91 to 4.99, while a higher pH range from 5.15 to 5.36 was found in samples dried using HAD and HAD + EAR. This difference could be explained by several factors, including exposure to drying duration, temperature, RH of hot air, and moisture migration rate. It might be caused by a reduction in dissociation of the organic acids during the prolonged drying duration of HAD and HAD + EAR. A previous study also proved that a shorter drying duration could lead to a low pH value [63]. By contrast, a significant increase in titratable acidity values was observed in chips dried using RFHAD (1.17 mg citric acid/100 g) and RFHAD + EAR (1.12 mg citric acid/100 g) compared to chips produced using the HAD (0.83 mg citric acid/100 g) and HAD + EAR (0.75 mg citric acid/100 g) methods (*p* < 0.05). A slow and gentle drying process enabled the evaporation of more acid, which might induce the higher pH (lower titratable acidity) of the dehydrated products [64]. This result was aligned with the finding published by Cui et al. [65]. In addition, chips dried using RFHAD + EAR and RFHAD had higher titratable acidity values, which agrees with the observation of a higher Vc retention of those two drying techniques in Table 3.

### 3.8. Enzyme Activity

The restructured chips underwent a variety of physicochemical alterations during dehydration process, which led to a notable influence on the enzyme activity of the dried samples. The impact of different dehydration techniques on the POD and PPO activities of the restructured chips is shown in Figure 7. The HAD + EAR samples had the highest PPO value (68.71 U·min^−1^g^−1^) among all dehydration techniques. For other dehydration techniques, the sequence was HAD (61.69 U·min^−1^g^−1^) > RFHAD + EAR (52.04 U·min^−1^g^−1^) > RFHAD (49.82 U·min^−1^g^−1^). A similar phenomenon was presented by Wang et al. [42], who observed that hazelnuts treated using RFHAD had a significantly lower PPO activity compared to that of HAD. This can be attributed to a higher temperature during RF processing, which could be beneficial for enhancing the inactivation effects on enzymes [66]. Similarity, the PPO values of chips dried using RFHAD (4.34 U·min^−1^g^−1^) and RFHAD + EAR (4.58 U·min^−1^g^−1^) were lower compared to those of samples dried using HAD (6.91 U·min^−1^g^−1^) and HAD + EAR (7.53 U·min^−1^g^−1^), which might explain the significantly higher L* values (lightness) of the RF-produced chips compared to those of the HAD-produced ones. The results were in agreement with the research conducted by Dag et al. [67]. Additionally, a notable positive correlation between the drying duration and PPO value was identified; thus, a short drying time would be preferred to preserve PPO activity in dehydrated chips.

### 3.9. Textural Property

Texture is an important property to evaluate the quality of processed products. Hardness is a key textural indicator which can be related to the force performed by mastication that happens during eating; a higher hardness value indicates more difficulty in chewing foodstuffs [68]. As reported in Figure 8, samples dried using HAD + EAR showed the largest hardness value (49.51 N), followed by HAD, RFHAD + EAR, and RFHAD, which were 40.93 N, 30.48 N, and 27.75 N, respectively. The measured hardness indexes of chips dehydrated using both RFHAD and RFHAD + EAR were significantly smaller than those of chips dried using HAD and HAD + EAR (*p* < 0.05). A probable reason for this could be that HAD caused severe physical and structural changes in the restructured chips during the drying process. A similar influence of the RF heating process on hardness was also stated in a previous report [69]. As the drying process continued for a longer duration, the hardness of samples increased. This observation was consistent with the study on the drying of mushroom by Xu et al. [70]. Meanwhile, this result also indicated that the RFHAD and RFHAD + EAR samples had a better texture (requiring a lower chewing force) compared to their counterparts. Furthermore, the RFHAD samples had a slightly lower hardness value than the samples dried using RFHAD + EAR, but the chips dried using RFHAD showed no statistical difference from those dried using RFHAD + EAR in terms of hardness force (*p* < 0.05).

### 3.10. Volume and Dimension Change

Volume change is one of the key quality attributes of dried products, as a loss in volume and change in shape may negatively affect consumers’ perception of dehydrated products [71]. This can also be reflected in physicochemical and structural changes that occur during drying. As shown in Table 5, processing methods had a huge impact on the SR, Dc, and Tc of dried restructured chips. The highest SR value was obtained from the HAD + EAR samples (22.26%), followed by HAD (20.85%) and RFHAD + EAR (19.13%), whereas the RFHAD samples showed the lowest SR value of 18.24%. Samples dried using RFHAD and RFHAD + EAR exhibited a significantly smaller SR than those prepared using HAD and HAD + EAR. This difference could be explained by the fact that the RFHAD and RFHAD + EAR process combined internal and external heating, enabling a faster drying rate. In contrast, the longer drying duration of HAD and HAD + EAR causes severe structural changes in the dried samples. This result was aligned with the investigation by Cao et al. [21], who found that RFHAD could reduce the shrinkage rate of dried tilapia. Moreover, shrinkage has a great impact on the textural characteristics of dehydrated products. It was also noted that the volume SR of the dehydrated chips had a positive correlation with the hardness index of the restructured chips produced by all four drying techniques.

The Dc of restructured chips processed using the four different drying techniques (HAD, HAD + EAR, RFHAD, and RFHAD + EAR) were 9.24%, 10.09%, 8.11%, and 8.27%, respectively. And the Tc of those four drying methods were 19.21%, 21.28%, 17.95%, and 18.34%, respectively. The change ratio in thickness after drying was much higher than the change ratio in diameter, which indicated that the main change occurred in thickness. The observed trend was aligned with the findings of Liu et al. [72] whose study compared different drying techniques on the dimension changes of asparagus snacks. Additionally, the ratios of change in diameter and thickness for chips produced using HAD/HAD + EAR were statistically higher in comparison with those observed in chips produced using RFHAD/RFHAD + EAR (*p* < 0.05). This could also be ascribed to the observed variation of SR caused by the different drying methods. The greater changes in shrinkage ratio corresponded to a greater ratio of changes in the thickness and diameter of the dried chips.

### 3.11. Moisture Content and Water Activity

The MC of restructured chips produced using different methods are shown in Table 6. The highest MC was determined in HAD + EAR chips (0.089 g/g d.m.), followed by HAD chips (0.081 g/g d.m.), RFHAD + EAR chips (0.076 g/g d.m.), and RFHAD samples (0.073 g/g d.m.). The MC of HAD + EAR chips was significantly different from that of the restructured chips processed using the RFHAD and RFHAD + EAR methods (*p* < 0.05). This could be due to the high RH of the exhaust air used in the HAD + EAR process. The longer processing duration might lead to a solid surface layer, which prevents the removal of moisture and results in dried products with a higher moisture content.

Water activity is a critical factor related to the quality and stability of food products; a lower water activity indicates a longer shelf life [73]. Table 6 also presents the water activity of chips produced using different drying techniques. The highest value of 0.245 was found in products dried using HAD + EAR, followed by HAD (0.222), RFHAD + EAR (0.201), and RFHAD (0.195), respectively. The tendency of water activity was aligned with the moisture levels of samples dried using different methods. Similar findings were stated by Moraga et al. [74] and Fazaeli et al. [75] for the drying of strawberry and black mulberry juice powder, respectively. Meanwhile, samples dried using RFHAD, with or without EAR, showed significantly lower water activity than their counterparts (*p* < 0.05). This could be ascribed to a greater reduction in water content achieved by means of volumetric RF drying. Similarly, Gong et al. [20] found that RFHAD-dehydrated carrots exhibited a lower water activity in comparison with HAD- and vacuum-freeze-dried samples.

### 3.12. Moisture Absorption Isotherms

The knowledge of the moisture absorption properties of foodstuffs is very important in food processing, especially as a means of quantitative methodology for predicting the shelf life of dehydrated foodstuffs [76]. Relative humidity and temperature are two major factors affecting product quality during storage. The strong moisture adsorption ability exhibits an adverse impact on the storage stability of dehydrated products, which potentially worsens product quality in an environment with high relative humidity [77]. The moisture sorption kinetics curves (RH: 15% to 75%, temperature: 25 °C) of products under four different drying processes are presented in Figure 9. A gradual increase in dry matter of all products was found under an RH of 55%. After storing the dried chips in the system at an RH of 55% for 72 h, the RFHAD samples displayed the highest moisture absorption capacity (weight change of 13.55%, d.m.), followed by HAD (11.05%, d.m.), RFHAD + EAR (10.34%, d.m.), and HAD + EAR (7.76%, d.m.). The moisture absorption capacities of RFHAD and HAD samples were higher than those of the samples dried using RFHAD and HAD with EAR. This might be attributed to the use of exhaust air with higher moisture which formed a solid surface layer on dried samples; this layer acted as a barrier to decrease the absorption capability. A previous study showed that brown rice dried using RFHAD exhibited a greater degree of water adsorption in comparison with that of HA samples [78]. Started from an RH of 55% onwards, all dried samples absorbed moisture and showed a rapid change in dry matter in a short time. At 25 °C, restructured chips dried using different methods had a weight change of 5.2~6.4% in the RH range from 35% to 55%, whereas in the RH range from 55% to 75%, the corresponding dry matter changes were dramatically increased by 9.8~17.8%. This revealed that moisture sorption at equilibrium under a higher RH level was less (or not) affected by the structure [79].

Figure 10 shows the moisture sorption isotherms (at 25 °C) of the restructured chips produced using four drying methods. It was perceived that within a low water activity range below 0.55, the moisture adsorption capacity of the dehydrated samples was low, which showed that the force between adsorbent (moisture in the storage environment) and adsorbate (restructured chips) was relatively weak. Consequently, the absorption capacity of the restructured chips increased following the increment of their water activity, indicating a classical porous filling behavior [77]. Moreover, when the water activity was lower than 0.55 (the RH of the storage environment was lower than 55%), the restructured chips processed using HAD + EAR showed the lowest moisture absorption capacity, followed by the RFHAD + EAR, HAD, and RFHAD samples. This observation might be explained by the following. (1) The use of EAR increased the humidity of exhaust air which might create a solid surface layer with fewer/smaller cracks on the dried samples, resulting in a weaker moisture absorption capacity under low RH conditions. Similar results were noted by Ogawa et al. [80]. (2) In some cases, severe shrinkage occurring during drying increased the ability to delay the absorption of liquids in dried products [81]. In addition, these experimental results might indicate the difficulty in controlling the quality of dried chips under conditions of an RH higher than 55% and at 25 °C.

The GAB model is widely used in the food sector due to its versatility and applicability across a broad range of water activity (0.05 < $a_{w}$ < 0.9) [34]. As presented in Table 7, the GAB model demonstrated a good fit for the data of moisture sorption isotherm curves of the dried chips, with the coefficient of determination (R^2^) being higher than 0.94. On the basis of Farahnaky et al. [82], the C values obtained from the GAB model for the moisture sorption isotherm curves of the dried chips were greater than 2, indicating a sigmoid Type II isotherm. Most importantly, the GAB equation also allowed the determination of M_m_ values for samples produced using different drying techniques, ranging from 0.0931 to 0.1012 (g/g d.m.). These M_m_ values were aligned with the values (0.0355 to 0.1556 g/g d.m.) reported in previous studies for starch-based cookies and biscuits [83,84]. The highest M_m_ value (0.1012 g/g d.m.) was found in the RFHAD samples, followed by the RFHAD + EAR samples (0.0975 g/g d.m.) and HAD samples (0.0968 g/g d.m.), whereas the HAD + EAR samples obtained the lowest M_m_ value of 0.0931 (g/g d.m.). These differences in M_m_ values could result from structural and physicochemical composition (such as polar groups, hydrophilic groups, and others) changes under different drying processes [85,86]. Overall, the application of EAR in the RFHAD and HAD methods resulted in the reduced M_m_ values of the dehydrated chips, potentially contributing to the improvement of storage stability for those products.

### 3.13. Principal Component Analysis (PCA) and Correlation Analysis

As shown in Figure 11a, the contribution of 1st principal component (PC1) = 86. 9%, and the 2nd principal component (PC2) = 6.5%. The PC1 and PC2 were accumulated for 93.4% of the total variance, which indicated that most determined data were representative and interpretable. Drying techniques situated in different quadrants on the biplot presented significant differences, and were further classified into two groups corresponding to their similarities. Therefore, chips produced using RFHAD and RFHAD + EAR were well discriminated from each other (based on PC1) and from samples dried using HAD and HAD + EAR (based on PC1). Meanwhile, the RFHAD and RFHAD + EAR group were located on the negative coordinate of PC1, and were highly correlated with D_eff_, TPC, DPPH, FRAP, Vc, TFC, and ABTS. To further interpret the findings obtained from PCA, the correlation analysis between different parameters is presented in Figure 11b. PC1 had a high correlation with D_eff_ (0.94), Vc (−0.99), TPC (0.85), TFC (0.67), DPPH (0.88), FRAP (0.95), ABTS (−0.72), and chlorophyll content (0.51). Meanwhile, PC2 clearly showed a negative correlation with TEC (−0.59) and drying time (−0.80).

As shown in Figure 11b, the results of CA were also plotted to reveal the specific correlation between the quality parameters of the restructured chips dehydrated using different techniques. Firstly, the drying time demonstrated significant negative correlations with chlorophyll (−0.72), Vc (−0.88), TFC (−0.73), and TPC (−0.72). This result indicated that a short drying time could effectively preserve a higher amount of bioactive components, which is in accordance with the results of bioactive component retention shown in Figure 3 and Table 3. Given that the values of DPPH, ATBS, and FRAP activities were strongly negatively correlated with color difference (ΔE), improving DPPH, ATBS, and FRAP activities might lead to a reduced ΔE. Additionally, it was observed that the bioactive components (Vc, TFC, and TPC) were strongly positively correlated with DPPH (0.84 to 0.97), ATBS (0.88 to 0.93), and FRAP (0.88 to 0.94). Similarly, Xu et al. [87] found that the decrease in antioxidant capacities could be attributed to the degradation of vitamin C, flavonoids, and phenolic compounds.

In general, the PCA and CA results demonstrated that RFHAD and RFHAD + EAR could be promising techniques for dehydrating restructured chips, because these two techniques were found to preserve the TPC, TFC, Vc, and antioxidant properties (DPPH, ABTS, and FRAP) of the restructured chips, while also improving drying efficiency (higher D_eff_ value and shorter drying duration) as well as reducing energy consumption.

## 4. Conclusions

The effects of RFHAD and HAD with EAR on the drying characteristics, D_eff_, SEC, TEC, moisture absorption isotherms, as well as selected key quality characteristics of restructured bitter melon and apple chips were investigated. According to the experimental results, RFHAD and RFHAD + EAR reduced the drying duration by 31~39% in comparison with HAD and HAD + EAR. The highest D_eff_ values obtained from RFHAD and RFHAD + EAR were 6.062 × 10^−9^ to 6.889 × 10^−9^ m^2^/s, while the lowest SEC values ranged from 301.57 to 328.79 kW·h/kg. With regard to product quality, samples dried using RFHAD and RFHAD + EAR exhibited better qualities (such as a lower color difference of 5.41~6.52, a lower shrinkage ratio of 18.24~19.13, a better antioxidant capacity, a higher vitamin C, total flavonoid, and total phenolic content, a lower POD activity of 4.34~4.58 U·min^−1^g^−1^, smaller diameter and thickness changes, and a lower hardness of 27.75~30.48 N) compared to those of samples dried using HAD and HAD + EAR. Furthermore, the assistance of EAR on RFHAD achieved the highest TEC reduction of 12.4%, and a better anti-absorption capacity, without compromising on most quality attributes of the dried chips. Hence, this novel RF drying concept is recommended for the industrial processing of several food products. Future investigations will be considered for pilot-scale applications in high-moisture foodstuffs, by using the RFHAD technique to improve product quality and drying efficiency.

## Figures and Tables

**Figure 2 foods-13-00197-f002:**
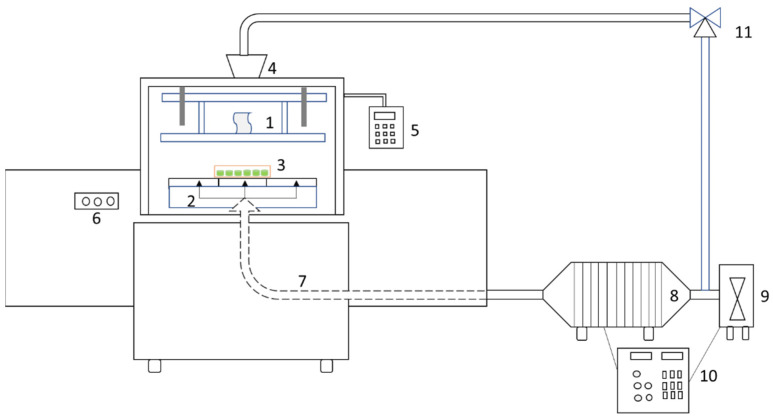
Simplified schematic of radio frequency heater assisted with hot air system: 1, top electrode; 2, bottom electrode; 3, samples; 4, exhaust air fan; 5, electric controller of RF heater; 6, safety switches; 7, hot air tube and distributors; 8, electric heater; 9, fresh air fan; 10, electric controller of hot air system; 11, three-way valve.

**Figure 3 foods-13-00197-f003:**
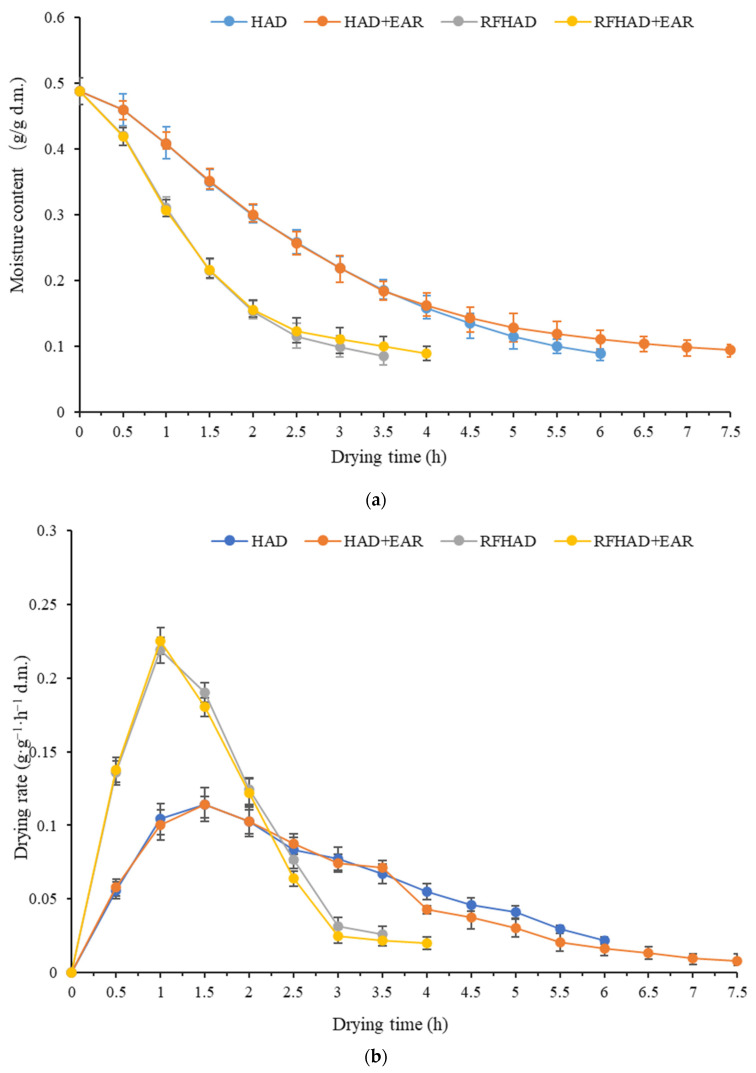
Moisture content curve (**a**) and drying rate curve (**b**) of the restructured bitter melon and apple chips processed using different drying methods.

**Figure 4 foods-13-00197-f004:**
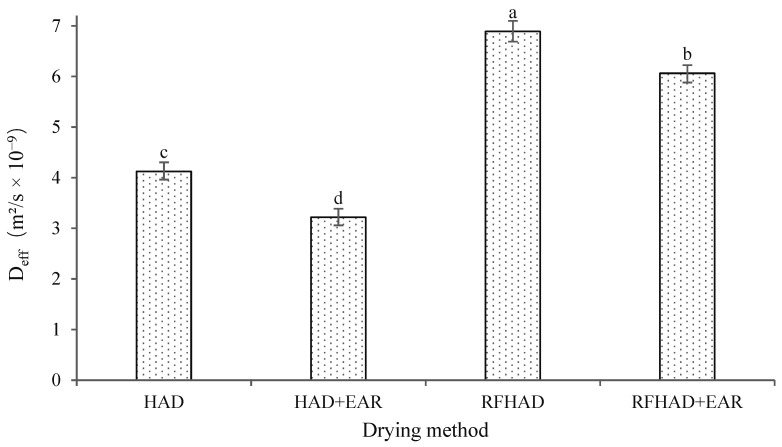
Effect of different drying methods on the effective moisture diffusivity (D_eff_) of the restructured bitter melon and apple chips. Values with different letters in each column are considered significantly different (*p* < 0.05).

**Figure 5 foods-13-00197-f005:**
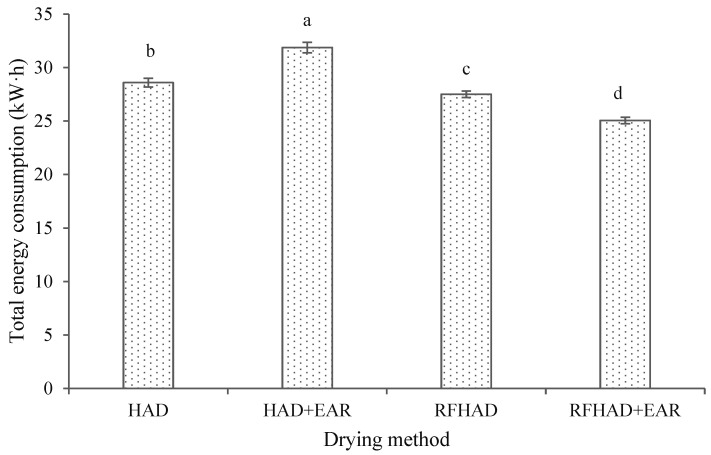
Effect of different drying methods on the energy consumption of the restructured bitter melon and apple chips. Values with different letters in each column are considered significantly different (*p* < 0.05).

**Figure 6 foods-13-00197-f006:**
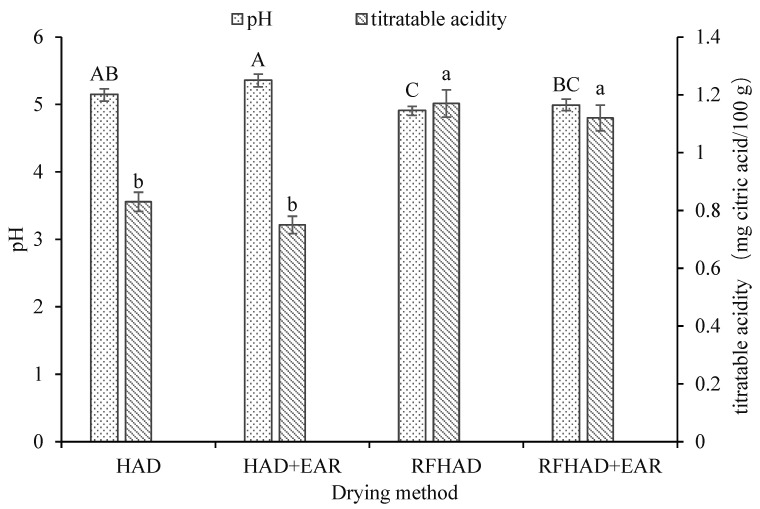
Effect of different drying methods on the pH and titratable acidity of the restructured bitter melon and apple chips. Values with different letters in each column are considered significantly different (*p* < 0.05).

**Figure 7 foods-13-00197-f007:**
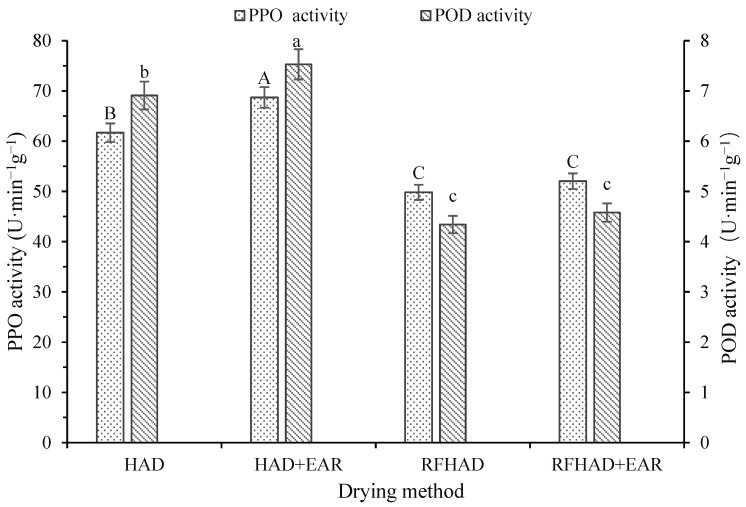
Effect of different drying methods on the PPO and POD activity of the restructured bitter melon and apple chips. Values with different letters in each column are considered significantly different (*p* < 0.05).

**Figure 8 foods-13-00197-f008:**
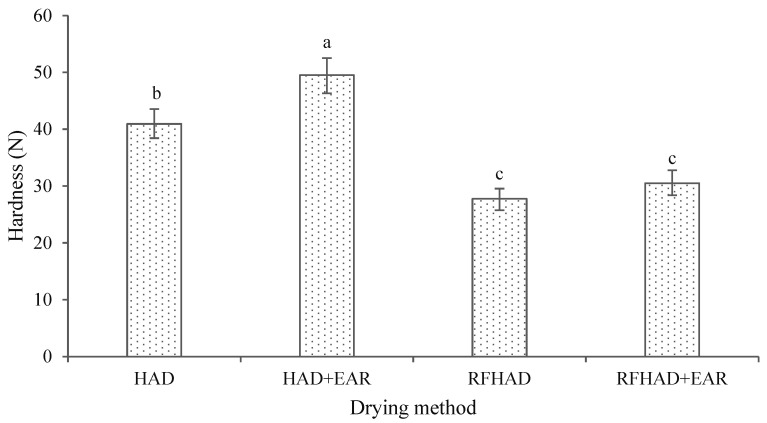
Effect of different drying methods on the hardness of the restructured bitter melon and apple chips. Values with different letters in each column are considered significantly different (*p* < 0.05).

**Figure 9 foods-13-00197-f009:**
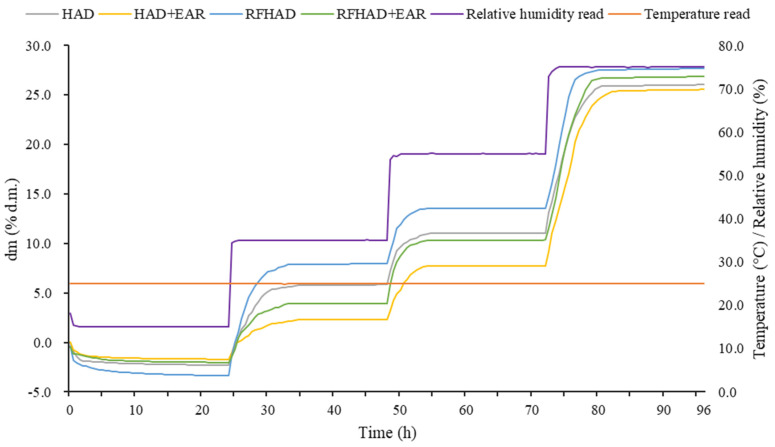
Moisture sorption kinetics curves of the restructured bitter melon and apple chips under different drying methods.

**Figure 10 foods-13-00197-f010:**
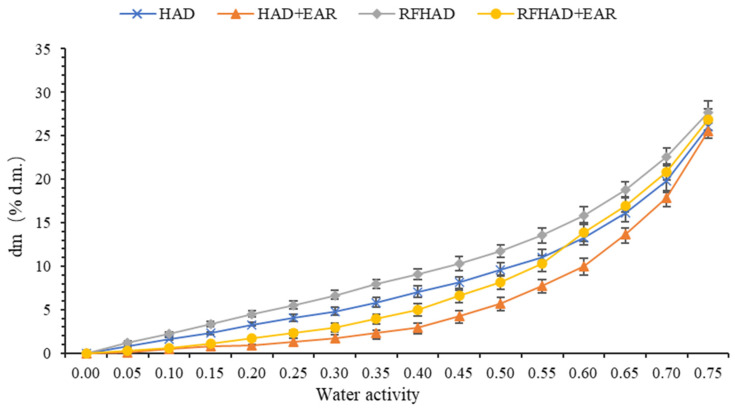
Moisture sorption isotherms of the restructured bitter melon and apple chips processed using different drying methods.

**Figure 11 foods-13-00197-f011:**
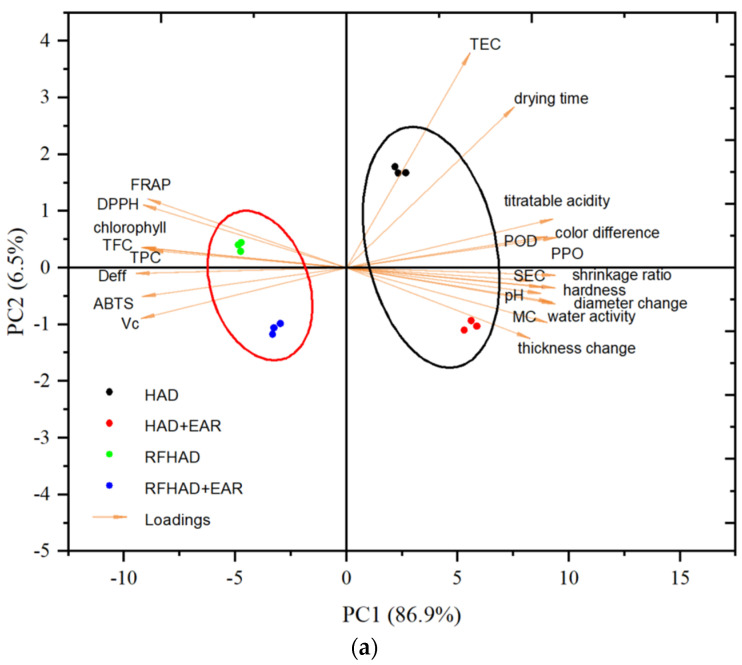
Principal component analysis (**a**) and correlation analysis heat map (**b**) of the restructured bitter melon and apple chips processed using different drying methods.

**Table 2 foods-13-00197-t002:** Changes in color (L*, a*, b*, C, H) and color difference (ΔE) of the restructured chips produced using different drying methods.

Samples	Picture	L*	a*	b*	C	H	ΔE
Fresh	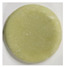	70.17 ± 0.43 ^a^	−5.34 ± 0.28 ^d^	16.97 ± 0.32 ^c^	17.73 ± 0.83 ^c^	107.47 ± 0.14 ^a^	-
HAD	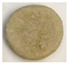	56.40 ± 1.09 ^c^	−0.20 ± 0.69 ^b^	19.53 ± 0.61 ^b^	19.66 ± 0.29 ^b^	89.62 ± 0.23 ^d^	14.87 ± 0.83 ^b^
HAD + EAR	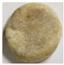	54.75 ± 1.22 ^c^	1.01 ± 0.97 ^a^	21.86 ± 0.74 ^a^	21.80 ± 0.33 ^a^	87.35 ± 0.29 ^d^	17.37 ± 0.99 ^a^
RFHAD	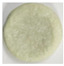	65.63 ± 0.79 ^b^	−2.63 ± 0.53 ^c^	17.39 ± 0.49 ^c^	17.92 ± 0.27 ^bc^	98.61 ± 0.17 ^b^	5.41 ± 0.57 ^c^
RFHAD + EAR	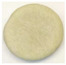	64.82 ± 0.95 ^b^	−1.76 ± 0.88 ^c^	18.06 ± 0.66 ^d^	18.25 ± 0.26 ^b^	95.57 ± 0.20 ^c^	6.52 ± 0.68 ^c^

Values are means ± SD of three replicates. Values with different letters in each column are considered significantly different (*p* < 0.05).

**Table 3 foods-13-00197-t003:** Effect of different drying methods on chlorophyll, Vc, TPC, and TFC of the restructured bitter melon and apple chips.

Drying Methods	Chlorophyll (mg/100 g d.m.)	Vc (mg/100 g d.m.)	TPC (mg GAE/100 g d.m.)	TFC (mg·RE/g d.m.)
HAD	15.64 ± 1.37 ^c^	5.07 ± 0.45 ^b^	8.18 ± 0.29 ^b^	2.65 ± 0.23 ^bc^
HAD + EAR	12.79 ± 1.48 ^d^	4.22 ± 0.53 ^b^	6.70 ± 0.27 ^c^	2.32 ± 0.29 ^c^
RFHAD	28.17 ± 1.21 ^a^	8.37 ± 0.39 ^a^	10.69 ± 0.34 ^a^	3.67 ± 0.28 ^a^
RFHAD + EAR	20.94 ± 1.29 ^b^	8.58 ± 0.36 ^a^	10.22 ± 0.40 ^a^	3.10 ± 0.31 ^ab^

Values are means ± SD of three replicates. Values with different letters in each column are considered significantly different (*p* < 0.05).

**Table 4 foods-13-00197-t004:** Effect of different drying methods on the antioxidant capacities of the restructured bitter melon and apple chips.

Drying Methods	DPPH (μmol TE/g d.m.)	ABTS (μmol TE/g d.m.)	FRAP (μmol TE/g d.m.)
HAD	7.52 ± 0.96 ^bc^	44.97 ± 1.85 ^c^	29.46 ± 1.47 ^c^
HAD + EAR	6.33 ± 1.04 ^c^	42.10 ± 2.36 ^c^	20.55 ± 1.83 ^d^
RFHAD	9.46 ± 0.72 ^a^	58.11 ± 1.73 ^a^	38.92 ± 1.35 ^a^
RFHAD + EAR	8.05 ± 0.81 ^ab^	54.27 ± 1.54 ^b^	34.04 ± 1.58 ^b^

Values are means ± SD of three replicates. Values with different letters in each column are considered significantly different (*p* < 0.05).

**Table 5 foods-13-00197-t005:** Effect of different drying methods on the shrinkage/dimension change of the restructured bitter melon and apple chips.

Drying Methods	SR (%)	D_C_ (%)	T_C_ (%)
HAD	20.85 ± 0.76 ^b^	9.24 ± 0.68 ^a^	19.21 ± 0.78 ^ab^
HAD + EAR	22.56 ± 0.83 ^a^	10.09 ± 0.75 ^a^	21.18 ± 1.25 ^a^
RFHAD	18.24 ± 0.52 ^c^	8.11 ± 0.49 ^b^	17.95 ± 0.59 ^b^
RFHAD + EAR	19.13 ± 0.59 ^c^	8.27 ± 0.54 ^b^	18.34 ± 0.66 ^b^

Values are means ± SD (*n* = 10 for shrinkage ratio; *n* = 8 for diameter change and thickness change). Values with different letters in each column are considered significantly different (*p* < 0.05).

**Table 6 foods-13-00197-t006:** Moisture content and water activity of the restructured chips dried using different drying methods.

Drying Methods	Moisture Content (g/g d.m.)	Water Activity
HAD	0.081 ± 0.07 ^ab^	0.222 ± 0.09 ^b^
HAD + EAR	0.089 ± 0.09 ^a^	0.245 ± 0.12 ^a^
RFHAD	0.073 ± 0.04 ^b^	0.195 ± 0.06 ^c^
RFHAD + EAR	0.076 ± 0.05 ^b^	0.201 ± 0.07 ^c^

Values are means ± SD of three replicates. Values with different letters in each column are considered significantly different (*p* < 0.05).

**Table 7 foods-13-00197-t007:** GAB model parameters of the restructured bitter melon and apple chips produced using different drying methods.

Drying Methods	M_m_ (g/g d.m.)	K	C	R^2^
HAD	0.0968	0.7287	5.1062	0.9753
HAD + EAR	0.0931	0.7653	4.6847	0.9698
RFHAD	0.1012	0.6949	5.9715	0.9642
RFHAD + EAR	0.0975	0.7382	5.3621	0.9480

R^2^, coefficient of determination.

## Data Availability

Data is contained within the article.

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
