# Peer review of "A High-Efficiency Radio-Frequency-Assisted Hot-Air Drying Method for the Production of Restructured Bitter Melon and Apple Chips"

_foods, 2024, doi:10.3390/foods13020197_

Round 1
Reviewer 1 Report
Comments and Suggestions for Authors
The manuscript presents interesting results from both a scientific and practical point of view. However this manuscript requires major revision due to some shortcomings.
Comments that should be considered to make the manuscript suitable for publication:
L 98. It is not clear whether the procedure for producing restructured bitter melon and apple chips is the authors' original invention or whether it was adopted from another known process.
L 107. What tool was used to cut chips with a diameter of 10 mm?
L 156. How long did it take to determine the moisture content? Was the time too long considering that during drying the samples were taken out of the dryer every 15 min at the end of drying?
L 211. Check whether the diameter of the penetrometer was 50 mm. The diameter of chips was only 10 mm.
L 215. Provide a reference for the definition of crispiness. The slope of the line representing the relationship between force and deformation relates to elasticity rather than crispiness.
L 260. Presenting of average drying rate is not appropriate. Consider representing the decrease in drying rate versus time or versus moisture content of the dried product.
L 277. Please provide to the caption of Figure 3 a description of the letters next to the bars in the Anova context. The same regards captions of other figures.
L 297. The increased drying time of HAD+EAR compared to HAD needs to be explained.
L 400. Consider to explain the swelling effect by volumetric heating which is responsible for puffing effect while vacuum-microwave drying.
L 424. Moisture content should be almost the same for all dried samples accordingly to the methodological approach described in line 149. Wasn't the final moisture content influenced by the delay effect depended on the drying rate in connection with the moisture content determination time? The assumption is that the drying process should continue until the same moisture content is reached despite od the drying method. On the other hand, water activity depends directly on the moisture content.
Comments on the Quality of English LanguageThe English writing style should be improved.
Author Response
Responses to the reviewer 1:
The manuscript presents interesting results from both a scientific and practical point of view. However this manuscript requires major revision due to some shortcomings.
Comments that should be considered to make the manuscript suitable for publication:
L 98. It is not clear whether the procedure for producing restructured bitter melon and apple chips is the authors' original invention or whether it was adopted from another known process.
Response: Thank you for your question. This procedure for producing restructured bitter melon and apple chips is our original invention in order to develop a restructured chips to avoid the bitter taste and also provide a new healthy product to consumers.
L 107. What tool was used to cut chips with a diameter of 10 mm?
Response: Thank you for your question. We used a stainless-steel baking mold (with the inside diameter of 10 mm) to manually form the chips.
L 156. How long did it take to determine the moisture content? Was the time too long considering that during drying the samples were taken out of the dryer every 15 min at the end of drying?
Response: Thank you for your question. It takes maximumly 4 hrs. to measure the moisture content of samples, the measurement time also varies based on the moisture content of fresh/dried samples.
According to preliminary experimental trials, the weight loss variation of dried samples was less than 0.01 g between consecutive measurements of every 15 min at the end of drying.
L 211. Check whether the diameter of the penetrometer was 50 mm. The diameter of chips was only 10 mm.
Response: Thank you for your suggestion. We have corrected the mistake in the article. The P5 cylindrical probe with contact area of 19.63 mm2 and diameter of 5 mm.
L 215. Provide a reference for the definition of crispiness. The slope of the line representing the relationship between force and deformation relates to elasticity rather than crispiness.
Response: Thank you for your question. We have analyzed the hardness as texture profile instead of crispiness and rewritten this part in the revised manuscript.
L 260. Presenting of average drying rate is not appropriate. Consider representing the decrease in drying rate versus time or versus moisture content of the dried product.
Response: Thank you for your suggestion. We have rewritten this part in the revised article.
L 277. Please provide to the caption of Figure 3 a description of the letters next to the bars in the Anova context. The same regards captions of other figures.
Response: Thank you for your question. We have made corrections in the revised manuscript.
L 297. The increased drying time of HAD+EAR compared to HAD needs to be explained.
Response: Thank you for your suggestion. We have made explanations and provided relevant reference in this section.
L 400. Consider to explain the swelling effect by volumetric heating which is responsible for puffing effect while vacuum-microwave drying.
Response: Thank you for your suggestion. We have rewritten this part in the revised article.
L 424. Moisture content should be almost the same for all dried samples accordingly to the methodological approach described in line 149. Wasn't the final moisture content influenced by the delay effect depended on the drying rate in connection with the moisture content determination time? The assumption is that the drying process should continue until the same moisture content is reached despite od the drying method. On the other hand, water activity depends directly on the moisture content.
Response: Thank you for your question. The moisture content determination time takes maximumly 4 hrs, and the determination time also varies based on the moisture content of fresh/dried samples. Meanwhile, the weight loss variation of dried samples was less than 0.01 g between consecutive measurements of every 15 min at the end of drying based on our preliminary experimental tests.
In addition, drying methods and drying parameters also have huge impact on the moisture content of dried products. For example: Drying Processes : Mass loss (ML) was monitored until samples reached a constant mass value. Experimental results showed that operating temperature led to a faster decrease and lower values in the MC of dried samples. Hence, drying process will not always achieve same moisture content under different drying methods and process conditions.
Figure 1. Effect of drying temperature on mass loss (g) (a) and variation of moisture content (%) (b) of samples with time.
(refer to article : Gonçalves, E. M., Pereira, N., Silva, M., Alvarenga, N., Ramos, A. C., Alegria, C., & Abreu, M. (2023). Influence of Air-Drying Conditions on Quality, Bioactive Composition and Sensorial Attributes of Sweet Potato Chips. Foods, 12(6), 1198.)
The English writing style should be improved.
Response: Thank you very much for your suggestion. We have improved in the revised manuscript.
Reviewer 2 Report
Comments and Suggestions for Authors
The paper proposes the use of radio frequencies to assist the drying process and considers air recirculation as a measure to make the process more energy efficient.
The work is interesting, however, there are opportunities for improvement which are detailed below:
In l22 avoid the use of words like viz and etc.
The sentence in l50 could be expanded a bit to improve the flow of the paragraph.
The document would improve if the idea of the advantages and differences of MW with RF were developed a little more, also the information regarding the energy issues of each method could be reinforced or expanded and emphasis could be placed on air recirculation to decrease energy consumption.
in 2.1 It is necessary to inform about the varieties used and also about characteristics that allow standardizing and reproducing the experiments, such as calibers, categories, some maturity indicator, just to mention some of them.
Is the diagram/figure 1 really necessary?
The diagram in figure 2 can be improved.
Paragraph l131-133 can be improved to make it easier to read.
Avoid the use of etc in l132.
It is necessary to clarify and justify why half of the time l141-142 was used.
Improve the wording of l148 (other parameters), also if they made measurements over time they could include the drying kinetics at least.
Important: if all measurements were performed in triplicate, it is not necessary to mention it in the different analyses, but rather at the end of the methodology. Also, in the document it sometimes appears in three and in others 3 for the replicates.
In the color measurements it is necessary to inform which was the illuminant and the observer used.
It would be interesting to clarify the meaning and usefulness of measuring changes in dimensions and volume change, it would be sufficient to report only one of these.
In l242-243 it is not necessary to explain the detailed operation of the equipment.
When mentioning the use of the sps-toolbox, is there any version that should be reported?
l285 avoid using the first person plural "our".
In figure 4, what is the reason for the increase in energy consumption when the equipment is recycled? should it not be the other way around?
It would be convenient to improve the discussion on energy issues by comparing with the work of other authors.
In l383 they talk about porous structures, did they make microstructural observations that allow to make such an assertion?
The discussion on volume change and dimensions can be improved.
In the discussion of humidity it is necessary to develop what would be the importance and meaning of the values obtained in context.
If they used a model for the isotherms (Flory-Huggins model) shouldn't there be constants with physical significance in the context of drying that could be analyzed?
The conclusion could also contain information related to concrete industrial applications and scaling.
Author Response
Responses to the reviewer 2:
The paper proposes the use of radio frequencies to assist the drying process and considers air recirculation as a measure to make the process more energy efficient.
The work is interesting, however, there are opportunities for improvement which are detailed below:
In l22 avoid the use of words like viz and etc.
Response: Thank you very much for your suggestion. We have made correction in the article.
The sentence in l50 could be expanded a bit to improve the flow of the paragraph.
Response: Thank you very much for your suggestion. We have rewritten this part.
The document would improve if the idea of the advantages and differences of MW with RF were developed a little more, also the information regarding the energy issues of each method could be reinforced or expanded and emphasis could be placed on air recirculation to decrease energy consumption.
Response: Thank you very much for your suggestion. We have rewritten this part.
in 2.1 It is necessary to inform about the varieties used and also about characteristics that allow standardizing and reproducing the experiments, such as calibers, categories, some maturity indicator, just to mention some of them.
Response: Thank you very much for your suggestion. We have added the category, and maturity indicator for bitter melon and apple.
Is the diagram/figure 1 really necessary?
Response: Thank you for your suggestion. We have deleted figure 1 in the revised manuscript.
The diagram in figure 2 can be improved.
Response: Thank you for your suggestion. We have made improvement on this figure in the revised manuscript.
Paragraph l131-133 can be improved to make it easier to read.
Response: Thank you for your suggestion. We have rephrased this section.
Avoid the use of etc in l132.
Response: Thank you very much for your suggestion. We have made correction in the article.
It is necessary to clarify and justify why half of the time l141-142 was used.
Response: Thank you for your suggestion and question. We have made additional explanations and rewritten this part in the revised article.
Improve the wording of l148 (other parameters), also if they made measurements over time they could include the drying kinetics at least.
Response: Thank you very much for your suggestion. We have made correction in the article.
Important: if all measurements were performed in triplicate, it is not necessary to mention it in the different analyses, but rather at the end of the methodology. Also, in the document it sometimes appears in three and in others 3 for the replicates.
Response: Thank you very much for your suggestion. We have made correction in the revised manuscript.
In the color measurements it is necessary to inform which was the illuminant and the observer used.
Response: Thank you for your suggestion. We have added the information of the illuminant and the observer used for the color measurement in the revised article.
It would be interesting to clarify the meaning and usefulness of measuring changes in dimensions and volume change, it would be sufficient to report only one of these.
Response: Thank you for your suggestion. We have made improved it and provided relevant reference in the revised manuscript.
In l242-243 it is not necessary to explain the detailed operation of the equipment.
Response: Thank you for your suggestion. The setup (relative humidity, temperature, measurement cycle) of SPS system is very important to determine the moisture absorption isotherms. For example, if the measurement cycle is too short, some products cannot achieve the equilibrium even under low relative humidity (e.g., 15%) condition.
When mentioning the use of the sps-toolbox, is there any version that should be reported?
Response: Thank you very much for your suggestion. We have added the version of sps-toolbox software in the revised manuscript.
l285 avoid using the first person plural "our".
Response: Thank you for your suggestion. We have corrected the mistake in the article.
In figure 4, what is the reason for the increase in energy consumption when the equipment is recycled? should it not be the other way around?
It would be convenient to improve the discussion on energy issues by comparing with the work of other authors.
Response: Thank you for your suggestion. We have made explanations and provided relevant reference in this section.
In l383 they talk about porous structures, did they make microstructural observations that allow to make such an assertion?
Response: Thank you very much for your suggestion. We have rewritten this part.
The discussion on volume change and dimensions can be improved.
Response: Thank you for your suggestion. We have corrected a mistake of calculation formulas for both diameter change (Dc) and thickness change (Tc). And we also made improvements of discussion section in the revised manuscript.
In the discussion of humidity it is necessary to develop what would be the importance and meaning of the values obtained in context. If they used a model for the isotherms (Flory-Huggins model) shouldn't there be constants with physical significance in the context of drying that could be analyzed?
Response: Thank you for your suggestion and question. We have made additional explanations and rewritten this part in the revised article.
The conclusion could also contain information related to concrete industrial applications and scaling.
Response: Thank you very much for your suggestion. We have corrected this part.
Reviewer 3 Report
Comments and Suggestions for Authors
I have carefully reviewed the manuscript. I found this quite interesting, but I have major and unavoidable concerns about the article. Considering these significant issues, I am not recommending the publication of this article in its current form, hence, a major revision. To justify my decision, I have the following points.
Line 15
I suggest you first draft it like this: Background (2 to 3 sentences) Scope and approach (2 to 4 sentences) Key findings This should cover >60% of the abstract Conclusions (2 to 3 sentences) After you draft it like this, then you remove the sections (background, scope & approach, key findings and conclusions" and you merge all the sentences.
Line 30
Provide a list of abbreviations
Line 59
Also, talk about the limitation of hot-air drying that necessitates advanced drying methods.
Line 90
Provide the year and month that it was purchased.
Line 133
Why didn’t you study the drying kinetics with time and also calculate the moisture diffusivity?
Line 138
Why did you select these parameters?
Line 159
How did you measure the moisture equilibrium? This is very important. You should have calculated it using s using a Guggenheim–Anderson–DeBoer (GAB) model.
Line 162
Present the figure for moisture ratio with time and perform a mathematical modelling on that.
Line 169
Perform the following research.
Drying rate curve
Drying kinetics
Moisture diffusivity
Antioxidants (abts, dpph, orac)
Tfc
Ph and titrable acidity
Enzyme activity
Line 171
Calculate the specific energy consumption.
Line 174
Also, calculate the chroma and hue angle
Line 197
Talk more about the extraction of vitamin C. This is very important. How did you ensure that there was no interference?
Line 201
Talk about the extraction procedure for TPC.
Line 207
Rather, measure the texture profile analysis. For instance, Δ Hardness, Δ Springiness, Δ Chewiness, and Δ Resilience)
Line 251
Also, add Principal component analysis to improve the manuscript.
this will lead to a convincing conclusion. PCA reduces the matrix dimensionality and identifies specific correlation patterns while retaining most information.
Line 257, results and discussion
Talk more about the engineering and science behind your results. Critically discuss them
This section is redundant as the results are already summarized and analyzed. In the present form, this section is not adequate as the abstract of the work. Rewrite, please.
Grammar
In its current state, the level of English throughout your manuscript does not meet the journal's desired standard. There are some grammatical and spelling errors and full stops missing, as well as instances of badly worded/constructed sentences. Please check the manuscript and refine the language carefully. I suggest you ask several colleagues who are skilled authors to check the English before your submission.
Comments on the Quality of English Language
major revision
Author Response
Responses to the reviewer 3:
I have carefully reviewed the manuscript. I found this quite interesting, but I have major and unavoidable concerns about the article. Considering these significant issues, I am not recommending the publication of this article in its current form, hence, a major revision. To justify my decision, I have the following points.
Line 15 I suggest you first draft it like this: Background (2 to 3 sentences) Scope and approach (2 to 4 sentences) Key findings This should cover >60% of the abstract Conclusions (2 to 3 sentences) After you draft it like this, then you remove the sections (background, scope & approach, key findings and conclusions" and you merge all the sentences.
Response: Thank you very much for your suggestion. We have rewritten the abstract.
Line 30 Provide a list of abbreviations
Response: Thank you for your suggestion. As descripted in Instructions for Authors from the website of FOODs Journal: Abbreviations/Initialisms should be defined the first time they appear in each of three sections: the abstract; the main text; the first figure or table. When defined for the first time, the acronym/abbreviation/initialism should be added in parentheses after the written-out form.
Line 59 Also, talk about the limitation of hot-air drying that necessitates advanced drying methods.
Response: Thank you very much for your suggestion. We have provided additional limitation of hot-air drying in the revised manuscript.
Line 90 Provide the year and month that it was purchased.
Response: Thank you very much for your suggestion. We have added the category, and maturity indicator for bitter melon and apple.
Line 133 Why didn’t you study the drying kinetics with time and also calculate the moisture diffusivity?
Response: Thank you very much for your suggestion. We have made correction in the article.
Line 138 Why did you select these parameters?
Response: Thank you for your suggestion. We have made additional explanations to this section.
Line 159 How did you measure the moisture equilibrium? This is very important. You should have calculated it using s using a Guggenheim–Anderson–DeBoer (GAB) model.
Response: Thank you for your question. According to China National Standard (GB 5009.3-2016), it takes maximumly 4 hrs. to measure the equilibrium moisture content of samples, the measurement time also varies based on the moisture content of fresh/dried samples.
Line 162 Present the figure for moisture ratio with time and perform a mathematical modelling on that.
Response: Thank you very much for your suggestion. We have made improvements in the revised manuscript.
Line 169 Perform the following research: Drying rate curve, Drying kinetics, Moisture diffusivity, Antioxidants (abts, dpph, orac), Tfc, Ph and titrable acidity, Enzyme activity, Calculate the specific energy consumption, Also, calculate the chroma and hue angle
Response: Thank you for your suggestion. We have performed additional research based on your constructive comments (such as moisture ratio with time, specific energy consumption, chroma and hue angle) in the revised article.
Line 197 Talk more about the extraction of vitamin C. This is very important. How did you ensure that there was no interference?
Response: Thank you for your suggestion. We have provided additional details of extraction procedure for vitamin C.
Line 201 Talk about the extraction procedure for TPC.
Response: Thank you for your suggestion. We have provided additional details of extraction procedure for TPC.
Line 207 Rather, measure the texture profile analysis. For instance, Δ Hardness, Δ Springiness, Δ Chewiness, and Δ Resilience)
Response: Thank you for your question. We have analyzed the hardness as texture profile instead of crispiness and rewritten this part in the revised manuscript.
Line 251
Also, add Principal component analysis to improve the manuscript.
this will lead to a convincing conclusion. PCA reduces the matrix dimensionality and identifies specific correlation patterns while retaining most information.
Response: Thank you for your suggestion.
Line 257, results and discussion
Talk more about the engineering and science behind your results. Critically discuss them
This section is redundant as the results are already summarized and analyzed. In the present form, this section is not adequate as the abstract of the work. Rewrite, please.
Response: Thank you very much for your valuable suggestion. We have made many improvements and rephrased this section in the revised manuscript.
Grammar
In its current state, the level of English throughout your manuscript does not meet the journal's desired standard. There are some grammatical and spelling errors and full stops missing, as well as instances of badly worded/constructed sentences. Please check the manuscript and refine the language carefully. I suggest you ask several colleagues who are skilled authors to check the English before your submission.
Response: Thank you very much for your suggestion. We have rewritten the entire manuscript.
Round 2
Reviewer 2 Report
Comments and Suggestions for Authors
Dear Editor
The authors have made most of the requested changes and the quality of the document has improved considerably. As a reviewer I am satisfied with the current version of the manuscript and recommend that it be accepted for publication.
Author Response
Thank you very much!
Reviewer 3 Report
Comments and Suggestions for Authors
Instead of using an excuse to make an unscientific analysis, you should have done the experiments on
Drying rate curve, Drying kinetics,
Moisture diffusivity, Antioxidants (abts, dpph, orac)
Tfc Ph and titrable acidity
Enzyme activity
The authors just calculated chroma and specific energy consumption, which is very basic and this doesn’t require any experiment.
Besides, I asked you to perform a Principal component analysis to improve the manuscript. This will lead to a convincing conclusion. PCA reduces the matrix dimensionality and identifies specific correlation patterns while retaining most information.
Go and run all the experiments I stated above before this can be published in an impact factor >5.0.
I am under no obligation to teach you how to present research results. But I trust my judgment that your research needs to be published, g. Goo and do all those experiments.
As a reviewer, I must ensure that the manuscript data I agree to be published is true. Since the author did not make any revisions after the reviewers raised this problem. I believe I could question the authenticity of the author's data. The authors need to prove the authenticity of their experiment. Considering these significant issues, I am not recommending the article for publication.
Comments on the Quality of English Languagemajor revision
Author Response
Response: Thank you very much for your constructive and valuable suggestions!
We have conducted additional experiments and analysis based on your recommendations (all the revised contents are marked in red), and also performed the principal component analysis (PCA) and correlation analysis (CA) for data analysis in the revised manuscript.
Round 3
Reviewer 3 Report
Comments and Suggestions for Authors
This is absolutely a better manuscript than the previous one. I recommend the acceptance of the manuscript for publication. Thanks
Author Response
Thanks a lot for your kind recommendation of acceptance..